# Prior-based Noisy Text Data Filtering: Fast and Strong Alternative For Perplexity

## Abstract

As large language models (LLMs) are pretrained on massive web corpora, careful selection of data becomes essential to ensure effective and efficient learning. While perplexity (PPL)-based filtering has shown strong performance, it suffers from drawbacks: substantial time costs and inherent unreliability of the model when handling noisy or out-of-distribution samples. In this work, we propose a simple yet powerful alternative: a **prior-based data filtering** method that estimates token priors using corpus-level term frequency statistics, inspired by linguistic insights on word roles and lexical density. Our approach filters documents based on the mean and standard deviation of token priors, serving as a fast proxy to PPL while requiring no model inference. Despite its simplicity, the prior-based filter achieves the highest average performance across 21 downstream benchmarks, while reducing time cost by over **1000×** compared to PPL-based filtering. We further demonstrate its applicability to symbolic languages such as code and math, and its dynamic adaptability to multilingual corpora without supervision. The code is available online ([https://anonymous.4open.science/r/prior_filter-D88D](https://anonymous.4open.science/r/prior_filter-D88D)).

## 1 Introduction

Large Language Models (LLMs) have achieved impressive performance by training on massive datasets, with web text serving as a primary data source. As web content continues to grow indefinitely, it offers unlimited data for pretraining. However, two major challenges necessitate careful filtering steps: (1) Web data is so large that we need to choose efficiently to save computational resources, and (2) It contains a lot of noise, which can harm the model if not properly filtered.

To address this need, various data selection methods have been proposed. Early approaches relied on heuristic rules [26, 4], but more recent trends have shifted toward model-based techniques [36, 18]. These methods typically involve training a reference model on a target dataset and using it to identify desirable data. The model may perform binary classification [35] or compute similarity with the reference dataset [36]. Among these, using the perplexity (PPL) score from a reference model as a criterion of filtering is currently known to offer the best performance while maintaining a relatively simple implementation [3]. We provide a more detailed review of related work in §A.

However, PPL-based approaches come with the following inherent limitations. (1) *Time cost*: These methods require training a reference model, followed by inference of PPL over the whole corpus. Given that web-scale data can easily exceed trillions of documents and continues to grow, performing inference over the entire corpus becomes prohibitively expensive. (2) *Reliability*: LLMs often fail to accurately assess samples from distributions that is not seen while training, such as noisy data. As a result, generative perplexity may sometimes assign high scores to noisy or low-quality text [11, 34].

This issue might become more pronounced when using smaller models to reduce inference costs, further undermining reliability.

To address this limitation of the PPL-based approach, we introduce a prior-based data filtering method grounded in linguistic insights. Instead of computing the full conditional probability of each token in the data $p(x_i|x_{<i}) \propto p(x_{<i}|x_i)p(x_i)$ ($x_i$ is token of a data d), this method focuses solely on estimating the prior term $p(x_i)$ with statistical metric such as term-frequency. It is extremely simple and significantly faster (almost **0.1% time consumption** compared to PPL-based), while it achieves even better performance on downstream task benchmarks.

Interestingly, this method is inspired by traditional techniques used in deciphering ancient languages. The 8th-century linguist Al-Kindi first proposed that, in order to decipher an encrypted language, analyzing the frequency of its words provides a clue [1]. If some word appears with the highest frequency across multiple documents, it is likely to correspond to a function word, such as "is" or "a" in English. This indicates that term-frequency itself is a one-dimensional representation for the role of a word: high frequency maps to function words while relatively low frequency maps to content words (e.g., "US", "president"). Combining with another linguistic observation that well-formed sentences within a language tend to exhibit a consistent level of lexical density (i.e., ratio between function and content words) [13], we can determine outlier document simply by computing the mean and variance of its term frequencies: which we term **prior-based data filter**.

The prior-based filter exhibits intriguing and practical properties, which we demonstrate empirically. (1) The linguistic principles underlying the term-frequency hold not only for English but also for other natural languages (e.g., Chinese and French), even for symbolic languages (e.g., code, mathematics). (2) Only a small amount of Chinese data mixed into an English corpus may be noise and models can not learn patterns from it; however, as its amount increases, it becomes learnable by models. The prior-based filter is capable of automatically capturing this transition of learnability.

We demonstrate that models pretrained using the prior-based filter outperform models using the PPL-based filter, across 21 diverse downstream task benchmarks. Moreover, since token priors can be estimated from a relatively small corpus, the prior-based filter is approximately 1000 times faster, requiring only 0.25 hours compared to 216 GPU hours for PPL-based filtering on a 6B-token corpus.

Our contributions are as follows:

- We propose the prior-based filter as an approximate alternative to the PPL-based filter.

- We analyze the useful properties of the prior-based filter, including its efficiency and generalizability.

- Through extensive downstream benchmarks, we demonstrate that the prior-based filter is not only faster, but also outperforms the current state-of-the-art PPL-based filtering.

## 2  Prior is a one-dimensional representation for the role of a token

In this section, we first introduce PPL-based approach, which is the previous SOTA for data filtering. Then we define how to estimate the prior, a key component of PPL. We then analyze the linguistic properties and significance of the prior, to show its potential as an effective criterion for data filtering.

### 2.1  PPL-based approach and estimation of prior

The PPL-based filtering method is known as the most effective approach for filtering noise data from web text corpus for pretraining LLMs [3, 18]. For the filtering, first, a small reference model $\theta$ (an autoregressive transformer architecture of 137M parameters) is trained on the corpus $D$. The model then computes the PPL for each data point $d = (x_1, x_2, \ldots, x_N)$, where $x_i$ is the token at the $i^{th}$ position of a document, and $d \in D$. Then, d with PPL values farthest from the median are discarded. Here, the PPL is defined as follows:

$$\text{PPL}(\text{d}) = \left[ \prod_{i=1}^{N} p_\theta(x_i|x_{<i}) \right]^{\frac{1}{N}} \tag{1}$$

$p_\theta(x_i \mid x_{<i})$ is the conditional probability of token $x_i$ given its preceding context $x_{<i}$ under the model $\theta$, that can be decomposed into likelihood and prior as follows.

$$p_\theta(x_i \mid x_{<i}) \propto p_\theta(x_{<i} \mid x_i) \cdot p_\theta(x_i) \tag{2}$$

In this Bayesian formulation, the likelihood term $p_\theta(x_{<i} \mid x_i)$ captures the dependency between the token $x_i$ and its preceding context $x_{<i}$, indicating how well the token aligns with the surrounding text. In contrast, the **prior** term $p_\theta(x_i)$ represents the marginal probability of the token $x_i$, independent of its context.

**Estimation of prior**     Due to the independent property of prior, it is no longer necessary for a transformer model to learn the joint probability in order to estimate the prior. Therefore, in this work, we assume the prior $p_\theta(x)$ of a token $x$ is approximated by simple statistics (i.e., term-frequency) in a corpus $D$, estimated as follows: $p_{\text{prior}}(x) = \frac{f_D(x)}{\sum_{x' \in V} f_D(x')}$. Here, $f_D(x)$ is the number of occurrences of token $x$ in corpus $D$, $V$ is the vocabulary set.

## 2.2 Frequency analysis in linguistics

To justify the use of a token prior as a filtering criterion, we draw on linguistic insights that reveal its strong connection to lexical and syntactic structure. Linguistics offers two key insights related to term frequency, and by combining them, we can derive its potential utility as a data filtering criterion.

**(1) Term frequency is a 1-dimensional representation of a word's role:**  The 8th-century linguist Al-Kindi first proposed an idea that is still widely used today [1]: to decipher ancient or encrypted languages, analyzing the frequency of its words gives a clue. If some word appears with the highest frequency across multiple documents, it is likely to correspond to a **function word** (e.g., "is" or "a" in English) that serves grammatical roles. In contrast, **content words** which carry semantic meaning (e.g., "US", "president") tend to appear with relatively lower frequency. Therefore, frequency itself can serve as a basis for distinguishing between function words and content words. In other words, term frequency (i.e., prior) can be seen as a one-dimensional representation of a word's functional role. We analyze that this property partially stems from the next property.

**(2) Well-formed sentences typically exhibit a consistent range of lexical density:**  As **lexical density** is defined as the proportion of content words against function words, it is known that well-formed sentences in a language typically maintain a certain range of lexical density [13]. From this, we can infer that broken and ill-formed sentences will deviate significantly from this range to be outliers.

By combining these two properties, we can derive a principle for identifying ill-formed documents. First, we use the token prior as a one-dimensional representation to estimate whether each token functions more like a content or function word. Then, by assessing the overall composition of function and content words, we can determine whether the document is an outlier.

# 3 Prior-based data filtering

In this section, we present an explanation and analysis of the prior-based data filtering method. (1) We first analyze the token-level term frequencies, demonstrating that linguistic insights are applicable at the token level. (2) We then apply this principle to build our filtering method. (3) Lastly, we validate its feasibility by analyzing data samples filtered by our approach.

## 3.1 Analysis on the token prior

We first analyze the token-level term frequencies by calculating the token priors (with formulation in §2.1) on the Dolma dataset [31]. As we sort them by the logarithm (Figure 1), we can observe that the token priors distinctly fall into three clusters based on their height and slope, supporting the thesis that the token prior serves as a 1-dimensional representation for the token's role.

The three clusters in Figure 1 are as follows. (1) *High-prior zone*: a steep slope of high-prior tokens. We can observe that this zone mainly consists of function words (e.g., "the", "a", "is", "you"). (2) *Middle-prior zone*: As the priors in this zone have a similar range, they form a wide and gentle slope. This zone seems to mainly contain tokens for content words (e.g., "Phone", "shortcut", "tackles",

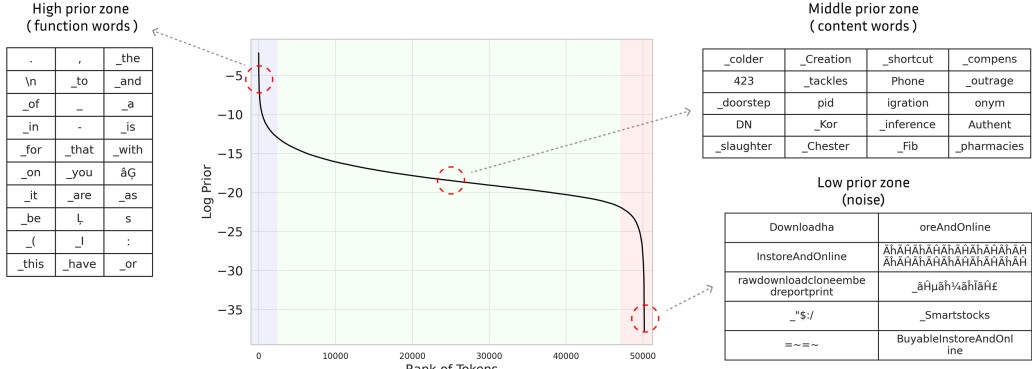

Figure 1: The line graph shows the logarithm of token priors (based on the GPT-2 tokenizer) computed from the Dolma dataset, sorted in descending order. The boxed regions highlight tokens from the top, middle, and bottom segments of the rank.

"doorstep"). (3) *Low-prior zone*: The frequency is extremely low, and the slope becomes steep again. This region is primarily composed of accidental noise tokens (e.g., "=≈ˀ'", "ÃĥÃĤÃĥÃĥÃĤÃĥ", "ĜãĤµãĥ¼ã"), including tokens from other language types that appear only a few times in the data (e.g., Chinese in English corpus).

## 3.2 Formulation of prior-based data filtering

We established two premises in §2.2: (1) A token's prior serves as a representation of its functional role, distinguishing function words from content words. (2) In a given language, a well-formed document typically maintains an average level of lexical density. By assessing the overall composition of function and content words, we can determine whether the document is an outlier.

However, since the prior is a continuous value rather than a discrete class label, we cannot directly compute the lexical ratio to assess the composition. Instead, we propose two alternative indicators to approximate the composition: the mean and standard deviation of token priors within a document.

**(1) Prior mean:** Since well-formed documents are clustered around a certain range of lexical density, the mean of token priors within such documents should also cluster around a certain value. **(2) Prior standard deviation:** Given that well-formed documents tend to exhibit a stable lexical density, the variance (or standard deviation) of token priors within a document should also cluster around a specific value. We denote these metrics as $\mu_{\mathtt{d}}$ and $\sigma_{\mathtt{d}}$ respectively, formulating as follows. Specifically, we define the prior mean with a logarithmic transformation, as it aligns with the prior term in the PPL formulation; this is discussed in more detail in §3.4.1:

$$\mu_{\mathtt{d}} = \mathbb{E}_{x_i \in \mathtt{d}}\left[\log p_{\text{prior}}(x_i)\right], \;\; \sigma_{\mathtt{d}} = \text{std}_{x_i \in \mathtt{d}}\left[p_{\text{prior}}(x_i)\right], \;\; \mathtt{d} \in D \tag{3}$$

As we assume that both $\mu_{\mathtt{d}}$ and $\sigma_{\mathtt{d}}$ of a well-formed document are clustered around certain central value, we define this central value as the median over the corpus $D$: $M_\mu = \text{median}_{\mathtt{d} \in D}(\mu_{\mathtt{d}}), M_\sigma = \text{median}_{\mathtt{d} \in D}(\sigma_{\mathtt{d}})$. The distance from the median is then used as a measure of outlierness. $\delta_\mu(\mathtt{d}) = |\mu_{\mathtt{d}} - M_\mu|, \delta_\sigma(\mathtt{d}) = |\sigma_{\mathtt{d}} - M_\sigma|$. To perform filtering, we discard the samples with the large $\delta$. The discarded portion is defined as the filtered set $F_\mu, F_\sigma$.

We analyze that the two criteria capture different aspects of the data. While $\delta_\mu$ captures the composition of tokens in a document, reflecting whether the document predominantly consists of high or low prior tokens, $\delta_\sigma$ reflects the distributional structure among tokens, indicating how uniformly or diversely the token priors are spread. This difference is also observed in the outlier samples.

## 3.3 Observation on distribution and outlier samples of $\mu_{\mathtt{d}}$ and $\sigma_{\mathtt{d}}$

We check whether the values of $\mu_{\mathtt{d}}$ and $\sigma_{\mathtt{d}}$ are clustered around central points, as hypothesized. For this, we randomly sample 600K examples from the Dolma dataset and compute $\mu_{\mathtt{d}}$ and $\sigma_{\mathtt{d}}$ for each $\mathtt{d}$. As shown in Figure 2, both values exhibit broad distributions centered around their respective

medians, with relatively small deviations. Notably, beyond a certain threshold, we observe sharp increases in deviation, forming clear outlier regions (highlighted by red dashed circles). Upon inspecting these outlier samples, we find that they primarily consist of noisy documents lacking meaningful information (boxes of Figure 2).

Figure 2: The line graph displays the values of $\mu_{\mathtt{d}}$ and $\sigma_{\mathtt{d}}$ computed from token priors in the Dolma dataset, sorted in descending order. Boxes are outlier samples from both distributions.

(a) $\mu_{\mathtt{d}}$ with outlier samples

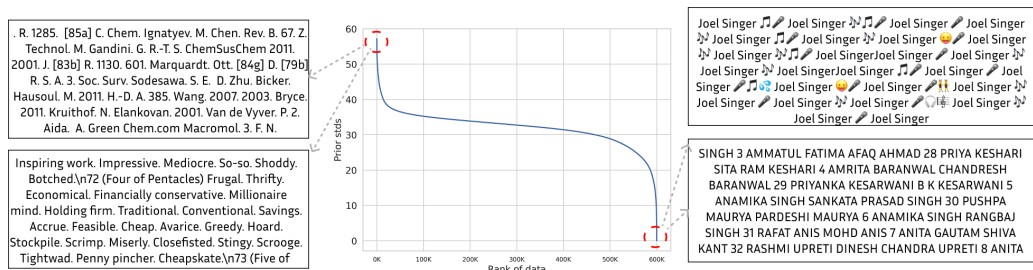

(b) $\sigma_{\mathtt{d}}$ with outlier samples

**Characteristics of outliers from each metric** We observe that the outliers for $\mu_{\mathtt{d}}$ and $\sigma_{\mathtt{d}}$ exhibit different characteristics. In the case $\mu_d$, the outliers tend to consist of tokens with either extremely high or extremely low prior values. For example, on the extreme-high side of the $\mu_d$ (left boxes of Figure 2a), documents mainly consist of line breaks ('\n') or space characters (' '), which is one of the tokens with the highest prior. On the extreme-low side (right boxes of Figure 2a), documents are often filled with non-English language or special characters.

Conversely, in the case of $\sigma_{\mathtt{d}}$, many outlier documents contain content-word tokens with middle-prior (boxes of Figure 2b). However, these words are arranged in unstructured ways, often appearing as a list of nouns without sentence structure. These differences arise because the $\mu_{\mathtt{d}}$ reflects the average composition of tokens in a document, whereas the $\sigma_{\mathtt{d}}$ captures the distributional pattern of those tokens. This suggests that both values should be used together for more effective data selection.

## 3.4 Properties of prior-based filter

### 3.4.1 Prior-based filter approximates PPL-based filter

The prior-based filter serves as an approximation to the PPL-based filter. We support this claim through both a formulation analysis and a statistical comparison of filtered data overlap.

$$\log \mathrm{PPL}(\mathtt{d}) \propto \underbrace{\sum_{i}^{N} \log p_\theta(x_{<i}|x_i)}_{\pi_{likelihood}} + \underbrace{\sum_{i}^{N} \log p_\theta(x_i)}_{\pi_{prior}} \tag{4}$$

First, the logarithmic form of PPL reveals that both the $\mu_{\mathtt{d}}$ and $\sigma_{\mathtt{d}}$ express two components of the PPL. (1) $\pi_{prior}$: The formulation of $\mu_{\mathtt{d}}$ in Equation 3 is exactly equivalent to the $\pi_{prior}$. (2) $\pi_{likelihood}$: as $\pi_{likelihood}$ captures the regularity of relationships among tokens within a document, $\sigma_{\mathtt{d}}$ similarly

reflects the regularity in distribution of token priors. This suggests that the two measures are weakly aligned. Taken together, combining $\mu_d$ and $\sigma_d$ can serve as a reasonable proxy for perplexity.

**Prior can be even better metric than PPL**  While $\sigma_d$ captures an approximation of the likelihood term, it is significantly more saturated than the actual likelihood, which can be considered a limitation. However, conversely, the inherent instability of the $\pi_{likelihood}$ (described as follows) poses a limitation for the PPL-based approach. (1) When the model is small, it struggles to accurately learn the likelihood [33]. (2) The model does not learn how to estimate likelihood for data from previously unseen distributions (mostly noisy data), which is not a problem for estimating only the prior. For this reason, previous studies have also reported that PPL often mistake repetitive or pattern-based noise as valid text [11]. Empirically, the model trained with the prior-based filter shows better downstream performance than the one trained with the PPL-based filter (§4).

Figure 3: Extreme outlier samples selected based on three criteria, ensuring that each sample comes from a distinct criterion: PPL, $\mu_d$, and $\sigma_d$.

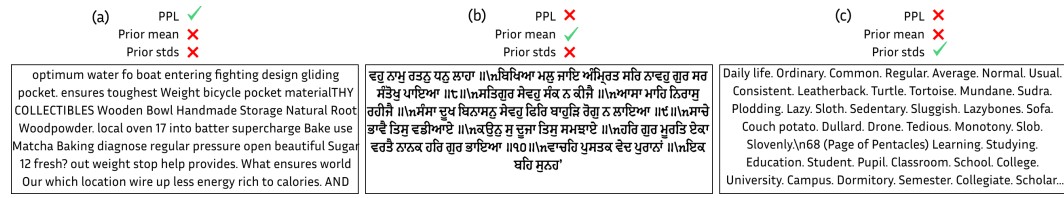

**Observation on filtered samples**  This characteristic of PPL is also observed in outlier samples. We investigate the most extreme outlier samples from each metric (PPL, $\mu_d$, $\sigma_d$), excluding their overlaps (Figure 3). As described in §3.3, outliers of $\mu_d$ tend to be filled with extremely low or high prior tokens (Figure 3b), while those of $\sigma_d$ often consist of content words but lack function words or valid sentence structure (Figure 3c). In outliers of PPL (Figure 3a), content and function words appear to be well-balanced, giving the surface impression of well-formed sentences, but upon closer reading, many of them turn out to be semantically meaningless. This may reveal both a strength and a weakness of the PPL metric: it effectively captures subtle irregularities within well-formed documents, but may fail to detect noise arising from entirely out-of-distribution samples.

**Statistical comparison**  To demonstrate that prior-based filtering approximates the PPL-based filter, we measure the overlap ratio of data filtered by each metric. We first randomly sample 600K examples from the Dolma dataset. Then, for each value ($\mu_d$, $\sigma_d$, PPL), we extract the data points whose percentile rank falls within the top or bottom $\frac{e}{2}\%$ (Figure 4). These are denoted as the filtered sets $F_\mu$, $F_\sigma$, and $F_{ppl}$, respectively. For each filtered set, we compute the overlap ratio with $F_{ppl}$, defined as $\frac{|F \cap F_{ppl}|}{|F_{ppl}|}$.

The results show a strong correlation: When filtering by the $e = 0.10$, nearly 50% of $F_\mu$ and $F_{ppl}$ overlap. We also find $F_\mu$ aligns more closely with $F_{ppl}$ than $F_\sigma$.

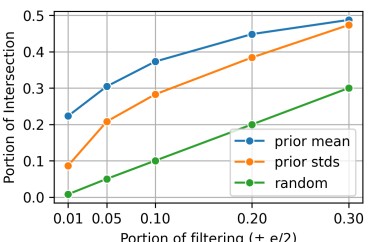

Figure 4: Overlap between outliers based on $\mu_d$ and $\sigma_d$ with those based on PPL, when filtering the top and bottom $\frac{e}{2}\%$ of samples (X-axis: $e$).

### 3.4.2  $\mu_d$ reflects language learnability in multi-lingual corpora

The prior mean value has the property of dynamically reflecting the learnability of a data cluster (i.e., language type), especially when multiple clusters with distinct characteristics are mixed. For example, consider a corpus primarily composed of English data with a small portion of Chinese data included. While the Chinese samples may contain meaningful content, if their quantity is too small, the model will fail to learn the language. In this case, Chinese data is no more than noise. However, once the volume of Chinese data increases sufficiently, the model becomes able to interpret the language, making it learnable and meaningful data.

The prior-based filter captures this dynamic behavior without any special tuning. As shown in Figure 3, prior mean values tend to classify non-English samples as noise when they are sparsely

mixed into English data. However, when the proportion of such data exceeds a certain threshold, the filter begins to treat them as valid language rather than noise.

To demonstrate this, we add a Chinese dataset (Wiki-ch)[1] to an English corpus (Dolma), with the Chinese data scaled to $a\%$ of the English corpus size. We then measure the percentage of added Chinese samples that fall into the outlier set (percentile rank falls within the top or bottom $10\%$). As shown in Figure 5, when the size of the Chinese data is only $1\%$ relative to the English data, nearly all of it is classified as noise. However, once its proportion exceeds $20\%$, the rate of being classified as outliers drops to a level comparable to random filtering ($10\%$, indicated by the red dashed line).

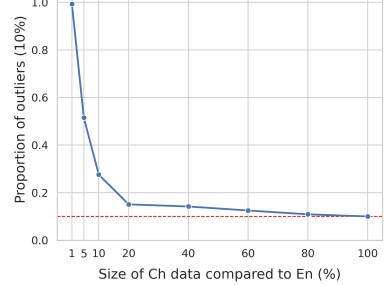

Figure 5: Proportion of Chinese data classified as outliers (Y-axis), after mixing Chinese and English data at a ratio of $a : 100$ ($a$ as X-axis). Outliers are the top and bottom 5% of $\mu_{\mathrm{d}}$.

This characteristic offers a major advantage over methods that require manually specifying a reference dataset (e.g., DSIR [36]). In DSIR, a human must decide whether to select English or Chinese data and then provide a suitable reference dataset accordingly. In contrast, the prior-based filter automatically determines whether a language should be filtered out based on its learnability.

### 3.4.3 Fast, scalable filtering using subsampled priors

One of the key advantages of the prior-based filter over model-based methods lies in its efficiency. Given the massive volume of new web data, which rapidly grows daily, training and inferring PPL value with a reference model can significantly amplify the time cost of filtering. In contrast, the prior-based filter only requires computing term frequencies and then calculating the mean and standard deviation of the priors.

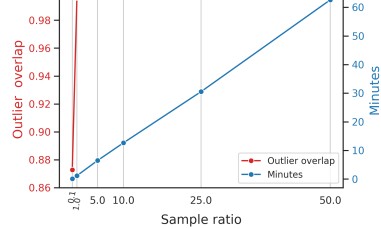

Figure 6: When token prior is computed with $b\%$ subset of Dolma (X-axis is $b$), the proportion of outliers overlapping with those from $b = 100$ is on the left Y-axis. The right Y-axis shows the computation time (in minutes) required to calculate the token prior at each $b$.

Remarkably, the already minimal computation time of the prior-based filter can be further reduced. For a 6B-token corpus, the entire process takes about 35 minutes on 40 CPUs (Intel Xeon Silver 4210R @ 2.40GHz), which consists of two stages: assessing the token prior, and computing $\mu_{\mathrm{d}}$ and $\sigma_d$. Among these, the most time-consuming step is the token prior assessing phase, which alone takes around 30 minutes.

This assessment time can be significantly reduced, as term-frequency estimates remain highly consistent even when calculated from a small subset of the data. To verify this, we sample $b\%$ of a 6B-token dataset to compute the token prior and then measure how much the resulting outlier set (top/bottom $10\%$) overlaps with the outlier set derived from the full corpus ($b = 100$). As shown in Figure 6, even with just $b = 1\%$, the extracted outliers are nearly identical to those from full corpus; requiring only about 70 seconds, or roughly one minute.

## 4 Experiment on downstream task

In this section, we evaluate the downstream task performance of models pretrained with different data filtering methods. Most training settings and hyperparameters follow those of [3]. We first conduct experiments on a natural language (specified to English) web corpus, Dolma [31]. This allows us to assess the effects on general language capabilities of a model (e.g., knowledge, language understanding, and symbolic understanding). To demonstrate that our prior-based method is applicable even to symbolic languages such as code and math, we also perform experiments on the Pile-github[2] dataset.

---

[1]https://www.kaggle.com/datasets/notoookay/chinese-wikipedia-2023
[2]https://www.kaggle.com/datasets/dschettler8845/the-pile-github-files-part-01

## 4.1 Experiment on natural language corpus and general ability

**Corpus setup**    Following [3], we mainly use Dolma [31] as a pre-training corpus for a testbed of filtering methods. Dolma is a large-scale, diverse web-text corpus, designed for training and evaluating LLMs. It contains noisy web data sources that support general language use ability, such as world knowledge, commonsense reasoning, and symbolic problem solving. This corpus is composed of multiple web-scale datasets, including Common Crawl, Reddit, Wikipedia, and Wikibooks[3], The Stack [15], C4 [26], PeS2o [30], Project Gutenberg

Table 1: Dolma v1.6 composition and its proportions based on token count.

| Source | Document type | portion |
|---|---|---|
| Common Crawl | web pages | 74.6% |
| The Stack | code | 13.4% |
| C4 | web page | 6.5% |
| Reddit | social media | 2.9% |
| PeS20 | educational papers | 2.3% |
| Project Gutenberg | books | 0.2% |
| Wikipedia, Wikibooks | encyclopedic | 0.1% |

[2] (see Table 1). Among these, Common Crawl accounts for the major portion (74.5%) of the corpus. This makes it a particularly suitable environment for evaluating filtering methods, as it contains a high proportion of noisy web content that must be thoroughly filtered, while a small but valuable subset (e.g., books and educational data) must be preserved. For testing under resource constraints, we select v1.6—a smaller subset with 6.3B tokens. We divide this into blocks (d) of 512 tokens, and select a subset of $N$ (3B) tokens for pretraining.

**Baseline setup**    When selecting a subset from Dolma, we follow the procedure defined by each method: (1) *no-filter*: Randomly selects $N$ without applying any filtering method. (2) *PPL-based*: Following the approach of [3] and §2.1, we first train a reference model (137M) on the random 3B tokens subset of dataset. We then compute the PPL score for each sample in the dataset. To obtain a final subset of size $N$, we discard samples with the highest and lowest PPL scores. (3) *DSIR*: Adopting the well-known method DSIR [36], we estimate n-gram frequency from the reference dataset (we choose Bookcorpus and Wiki-en) and compute importance weights. (4) *prior-based* (ours): As described in §3.2, we first estimate token priors using a 10% subset of the full corpus. Based on these priors, we compute $\mu_d$ and $\sigma_d$ (d $\in D$). We then discard samples with the highest $\delta_\mu$ and $\delta_\sigma$ values in the constraint of $|F_\mu| = |F_\sigma|$, until the volume of final subset $|F_\mu \cup F_\sigma|$ reaches $N$.

We use the GPT-2 architecture for pretraining, with large (1.5B) and small (137M) size models, using 8 GPUs (RTX A5000). Following [3], we set a max token length of 512, a global batch size 256, and a learning rate 2e-4, and train for 40K global steps (about 6B token duration). According to Ankner et al. [3], the relative performance trends observed at 40K steps are maintained in later training steps.

**Benchmark and evaluation setup**    The types and settings of downstream tasks follow those used in the [3], based on the MosaicML evaluation gauntlet [20]. Gauntlet includes tasks designed to assess five core capabilities: world knowledge, common sense reasoning, language understanding, symbolic problem solving, and reading comprehension. We normalize the accuracy of the individual task as $a_n = \frac{a_m - a_r}{1 - a_r}$, where $a_m$ is the accuracy of the model and $a_r$ is the expected accuracy of random guessing. We report the average normalized accuracy for each task, task category, and the average across all categories. Since some tasks are not proper for 1.5B models, we exclude benchmarks with average $a_n$ of baselines under 0.001. This results in a total of 21 benchmarks (details in §C)

**Results**    As described in Table 2, the results show that the model trained with *prior-based* filtering achieves the highest average performance, with extremely small time cost. Key observations are as follows: (1) *DSIR* outperforms *no-filter*, and *PPL-based* outperforms *DSIR*, which aligns with findings from previous research [3, 36]. (2) *Prior-based* filter approximates *PPL-based* filter in principle, but yields better downstream perfor-

Table 2: Performance and time cost (for filtering) of the baselines pre-trained on Dolma across 21 benchmarks. The average normalized accuracy is the average of all ability categories.

| | Time | Average normalized accuracy | World knowledge | Commonsense reasoning | Language understanding | Symbolic problem solving | Reading comprehension |
|---|---|---|---|---|---|---|---|
| **Large (1.5B) model** | | | | | | | |
| no-filter | - | 5.39 | 3.54 | 0.44 | 6.14 | **13.22** | 3.59 |
| DSIR | 4 hours | 7.09 | 4.65 | 6.84 | 7.31 | _12.67_ | **3.97** |
| PPL-based | 216 GPU hours | _7.65_ | 7.12 | 11.91 | _7.34_ | 7.91 | _3.96_ |
| Prior-based (ours) | 0.25 hours | **8.59** | 6.47 | _11.27_ | 10.31 | 11.13 | 3.79 |
| **Small (137M) model** | | | | | | | |
| no-filter | - | 4.68 | 3.59 | 1.81 | 1.47 | **12.83** | _3.70_ |
| DSIR | 4 hours | _5.23_ | 3.86 | 4.93 | 1.97 | _11.60_ | **3.80** |
| PPL-based | 216 GPU hours | 4.92 | _3.75_ | _6.53_ | _2.90_ | 7.84 | 3.58 |
| Prior-based (ours) | 0.25 hours | **6.26** | 3.10 | 9.13 | 4.22 | 11.21 | 3.66 |

mance. We analyze that this is because the PPL score depends on the model's likelihood, which can be unstable. On the other hand, the prior is based on simple word frequencies, so it gives a more stable and reliable signal. (3) Though the *prior-based* model outperforms the *PPL-based* model in downstream performance, the *prior-based* filtering requires significantly less processing time.

---

[3]https://commoncrawl.org/,  https://www.reddit.com/,  https://dumps.wikimedia.org/

PPL-based filtering takes 216 GPU hours to select a 3B token subset ($20 \times 8$ GPU hours of training the reference model, $7 \times 8$ GPU hours of PPL inference), while *prior-based* filtering takes only 15 minutes (6 minutes of assessing token prior, 6 minutes of calculating $\mu_d$ and $\sigma_d$ in $D$)—under 0.1% of the time spent for PPL. This demonstrates the superior scalability and efficiency of the *prior-based* approach.

(4) In symbolic problem solving, *PPL-based* filtering performs the worst, whereas *prior-based* filtering performs competitively with other baselines. This suggests that PPL fails to capture small and meaningful segments of different types of data, while *prior-based* filtering is more robust in preserving them. This is due to the property of $\mu_d$ that reflects the learnability of multiple language types (§3.4.2). (5) While *no-filter* performs poorly across most abilities, it shows the highest score in symbolic problem solving. This might be because small but meaningful portions of data (e.g., math or programming-related) are partially filtered out in other methods, but retained in the *no-filter*. For a *prior-based* filter, this issue can be handled by augmenting the small subset of the corpus for the targeted data type (i.e., datasets focused on coding or mathematics). This adjustment is straightforward and incurs minimal effort. (6) Across other skill categories, the *prior-based* method consistently outperforms other baselines or performs comparably to the best-performing one, resulting in the highest overall performance. (7) This trend remains consistent even for different-sized models.

## 4.2 Experiment on symbolic language corpus

We retain most of the settings from experiments of §4.1, including baselines and training configurations, but change the pretraining corpus to Pile-github. From the subset of 6B tokens, we extract a subset of 3B tokens with each filtering method. We exclude DSIR due to the difficulty of determining an appropriate reference dataset for Pile-github. This is also a critical limitation of the *DSIR*.

Pile-github mainly consists of code scripts, additionally containing a little mathematical data and natural language data. As it contains little information related to general language skills, such as world knowledge, we limit the evaluation only to 6 symbolic problem-solving benchmarks in gauntlet.

**Results** The observed results are as follows: (1) Consistent with the previous experiments, the *prior-based* method achieves the best performance with significantly less time than the *PPL-based* approach. (2) These findings suggest that our methods holds not only for natural languages (e.g., English, Chinese) but also for artificial symbolic languages (e.g., code, math). This means that well-formed data in a certain language type can be identified via *prior-based* statistics, regardless of language type. (3) Math-related benchmarks (BIG-bench elementary math QA, GSM8K, SVAMP) exhibit near-random performance across all baselines, likely because the Pile-github dataset consists predominantly of code scripts.

Table 3: Performance of the baselines pre-trained on Pile-github across 6 symbolic problem solving benchmarks

| | Time | Average normalized accuracy | BIG-bench cs algorithms | BIG-bench dyck languages | BIG-bench operators | BIG-bench elementary math QA | GSM8K | SVAMP |
|---|---|---|---|---|---|---|---|---|
| Large (1.5B) model | | | | | | | | |
| no-filter | - | 9.51 | 35.75 | 12.30 | 5.71 | 1.15 | **0.15** | **2.00** |
| PPL-based | 224 GPU hours | 11.21 | 37.42 | 20.60 | 7.14 | **2.09** | 0.00 | 0.00 |
| Prior-based (ours) | 0.26 hours | **12.03** | **38.86** | **21.30** | **9.04** | 1.17 | **0.15** | 1.67 |
| Small (137M) model | | | | | | | | |
| no-filter | - | 10.15 | 37.87 | **16.30** | 5.23 | 1.52 | 0.00 | 0.00 |
| PPL-based | 224 GPU hours | 9.82 | **40.45** | 14.10 | 1.42 | 2.61 | **0.07** | 0.33 |
| Prior-based (ours) | 0.26 hours | **12.19** | 40.22 | 16.00 | **7.14** | **3.08** | 0.00 | **6.66** |

# 5 Conclusion and limitation

We proposed a prior-based text data filtering method grounded in linguistic insight. The prior-based filter serves as an approximation of PPL-based methods, while achieving superior downstream performance and being over 1000× faster. Furthermore, it shows strong generalizability by performing effectively even on symbolic languages. This enables efficient filtering of rapidly growing web text data and provides a foundation for faster continual pretraining of LLMs.

However, since this method leverages linguistic properties, unlike other approaches such as PPL-based filtering or DSIR, it is less suited for extension to other modalities such as image data.

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

# A  Related works

In this section, we review previous works on web text data filtering for the pretraining of LLMs, and then more closely describe those sharing conceptual similarities with our proposed method.

**Rule-based**  Raw web-scraped data often contains a substantial amount of low-quality content, including documents with only space or machine-generated spam [16]. As a result, previous research has focused on effective filtering strategies. One of the most fundamental approaches is based on heuristic rules, such as retaining only English-language documents, removing samples that contain blocklisted words, or filtering out sentences that do not meet specific length criteria [4, 25]. However, such heuristic methods often fail to apply fine-grained filtering and risk discarding semantically valuable content inadvertently [9].

**Model-based**  More sophisticated approaches have been proposed that leverage the capabilities of deep neural networks, achieving superior performance compared to heuristic filtering. For instance, EL2N [24] ranks samples based on the L2 distance between a model's prediction and the ground truth, thereby identifying data points that are more important for learning. Similarly, memorization-based methods [5] assess how well a model memorizes token sequences within a document. Among these, [18, 3] demonstrated that using perplexity scores from a reference model to filter out both tails of the data distribution outperforms other techniques. In addition, DSIR [36] learns a bag-of-ngrams representation and uses n-gram similarity to perform data selection, which we discuss in the next section.

## A.1  DSIR

DSIR [36] assumes that a well-curated reference dataset consisting of high-quality, well-formed text is available (Wikipedia and Bookcorpus is used in the original work). The method is to evaluate the similarity of sample d in the raw dataset to this reference corpus, and uses it as the filtering criterion.

According to [36], the process for estimating this similarity proceeds as follows. Given a corpus $D$, each document $\mathtt{d} \in D$ is sliced into a sequence of $n$-grams. For example, if the text input is "Alice is eating", it forms the list [`Alice, is, eating, Alice is, is eating`]. These $n$-grams are then mapped to hash indices, which are subsequently grouped into $m$ hash buckets (with $m = 10000$). The resulting hash frequencies form an $m$-dimensional categorical distribution vector $\gamma \in \mathbb{R}^m$, referred to as the feature distribution $P$. Separate feature distributions $P_{raw}$ and $P_{ref}$ are computed for the reference dataset and the raw dataset, respectively (each denoted as $q$ and $p$ in the original paper).

From the feature distributions, we can derive feature extractors $P(\mathtt{d})$ as follows:

$$P(\mathtt{d}) = \prod_{j=1}^{m} \gamma[j]^{\mathtt{d}[j]} \tag{5}$$

$\mathtt{d}[j]$ indicates $j_{th}$ element of the sample $\mathtt{d}$. With this, we can calculate the importance weight for each data: $w(\mathtt{d}) = \frac{P_{ref}(\mathtt{d})}{P_{raw}(\mathtt{d})}$. The final selection is made by retaining those with the highest $w(\mathtt{d})$.

Here is a polished academic-style translation of your paragraph:

**Comparison with Our Method.**  If we set the $n$-gram size to $n = 1$ and let the number of hash buckets $m$ equal the vocabulary size, the DSIR feature distribution $P$ essentially becomes the token prior used in our work. Moreover, the computation of our $\mu_{\mathtt{d}}$ (the mean log prior of tokens in a document) is conceptually similar to DSIR's feature extraction process.

However, our approach differs in several important ways: (1) Unlike DSIR, which requires both the feature distribution of the raw and the reference dataset, our method relies solely on the raw dataset. This reduces the dependency and effort for a high-quality refined reference. In practice, obtaining a truly noise-free dataset is difficult, as corpora like Wikipedia or BookCorpus (used in DSIR) also have noise. Furthermore, for diverse domains (e.g., GitHub, Chinese corpora), DSIR demands a separate domain-specific reference corpus, which introduces additional overhead and subjectivity in selecting appropriate reference data.

(2) DSIR typically uses bigrams ($n = 2$), while our method is based on unigrams ($n = 1$). As a result, function words in DSIR are often tied to neighboring content words and rarely appear independently in the feature distribution, like in the example [`Alice, is, eating, Alice is, is eating`]. Consequently, DSIR's distribution tends to reflect the frequency of content words while neglecting the function words. This indicates a difference in the filtering principle from our approach.

## A.2   SCDP

SCDP (Swift Cross-Data Pruning) [21] is a method that selects data based on the multivariate median of TF-IDF (term frequency and inverse document frequency) representations. This method selects data that is most similar to the dominant topic frequently covered in the corpus.

To describe the method, first, a feature vector $\mathbf{t}_i = TF_i \odot IDF_i$ is computed for each $\mathtt{d} \in D$. And documents that are closest to the median (multivariate median) are selected.

Compared to our approach, SCDP differs in a fundamental way: whereas we compute token priors based on $TF \odot DF$, SCDP uses $TF \odot IDF$, which is the inverse way of reflecting $DF$. Because tokens with high document frequency receive lower $IDF$ scores, the function words are down-weighted or often entirely suppressed. As a result, SCDP's representation captures the frequency of content words only. This is in contrast to our method, which treats both function and content words as integral components of a document.

Such an approach leads to the following characteristics: (1) By eliminating the influence of function words, the method focuses on the composition of content words (i.e., topic), rather than on grammatical regularity. (2) Since selection is based on the median value, it favors documents that are closely related to one most frequent topic in the corpus.

This approach has a limitation in that the topic of the document does not necessarily correlate with its noise level. More specifically: (1) A corpus typically contains a diverse range of topics, some of which may be represented by only a small number of samples. If selection is based on topic similarity, informative but underrepresented data may be filtered out, even if it is not noisy. (2) Conversely, documents that align closely with the median topic can contain noise, while still being selected. For example, as exhibited in Figure 2b, certain web data consists of norm lists or repetitive content that may appear topically relevant but lack meaningful or well-structured information.

Due to these reasons, we argue that our approach is more optimal for identifying ill-formed, noise-heavy documents. This is because our method evaluates data based on whether the sentence is structurally well-formed, regardless of its topic.

# B   Details on experiments

Table 4: Benchmark performance of large (1.5B) models.

| Model | World knowledge | | | | Commonsense reasoning | | | Language understanding | | | |
| --- | --- | --- | --- | --- | --- | --- | --- | --- | --- | --- | --- |
| | ARC easy | BIG-bench wikidata | TriviaQA | MMLU | COPA | OBQA | PIQA | HellaSwag | LAMBADA | Winograd | Winogrande |
| no-filter | 8.25 | 2.81 | 0.40 | -0.42 | 0.31 | -4.00 | 15.34 | 1.30 | 6.68 | 12.82 | 3.71 |
| DSIR | 9.65 | 4.42 | 0.47 | -0.10 | 1.47 | 0.53 | 16.00 | 2.70 | 13.43 | 13.55 | -0.71 |
| PPL-based | 11.79 | 8.19 | 0.87 | 1.41 | 2.34 | 0.27 | 19.48 | 4.11 | 16.85 | 9.89 | -1.18 |
| Prior-based (ours) | 12.29 | 6.78 | 1.27 | 0.35 | 1.38 | -0.53 | 20.35 | 5.84 | 18.46 | 14.29 | 2.45 |

| Model | Symbolic problem solving | | | | | | Reading comprehension | | | |
| --- | --- | --- | --- | --- | --- | --- | --- | --- | --- | --- |
| | BIG-bench algorithms | BIG-bench dyck languages | BIG-bench elementary math QA | BIG-bench operators | GSM8K | SVAMP | LSAT-LR | LSAT-RC | SAT-English | CoQA |
| no-filter | 37.12 | 13.00 | 2.21 | 7.14 | 0.00 | 6.67 | 3.79 | 3.48 | 6.80 | 0.31 |
| DSIR | 39.92 | 13.70 | 2.70 | 5.71 | 0.15 | 1.33 | 3.79 | 4.48 | 6.15 | 1.47 |
| PPL-based | 25.23 | 0.60 | 3.27 | 7.14 | 0.68 | 3.33 | 3.53 | 4.48 | 5.50 | 2.34 |
| Prior-based (ours) | 33.03 | 11.50 | 3.75 | 5.71 | 0.23 | 1.67 | 3.01 | 3.98 | 6.80 | 1.38 |

Table 4 reports the performance of large (1.5B) models on Dolma across different filtering methods. As discussed above, the *prior-based* generally outperforms other baselines or performs comparably to the best baselines.

# C   Details on eenchmarks

Jha et al. [12] also use the MosaicML evaluation gauntlet to perform evaluations in their work. As such, with explicit permission from the authors, we reproduce their text describing the tasks and task categories in the evaluation gauntlet. The following is from Section D of their paper:

The **World Knowledge** category includes the following datasets:

- **ARC easy**: 2,376 easy four-choice multiple choice science questions drawn from grade 3-9 science exams. [7]
- **BIG-bench wikidata**: 20,321 questions regarding factual information pulled from Wikipedia. [32]
- **TriviaQA**: 3,000 question-answering dataset; clipped all answers to be at most 10 tokens long to improve speed. [14]
- **MMLU**: 14,042 four-choice multiple choice questions distributed across 57 categories. [10]

The **Commonsense Reasoning** category loosely assesses a model's ability to do basic reasoning tasks that require commonsense knowledge of objects, their properties and their behavior. It includes the following datasets:

- **COPA**: 100 cause/effect multiple choice questions. [28]
- **OBQA (OpenBook QA)**: 500 four-choice multiple choice questions that rely on basic physical and scientific intuition about common objects and entities. [19]
- **PIQA**: 1,838 commonsense physical intuition 2-choice multiple choice questions. [6]

**Language Understanding** tasks evaluate the model's ability to understand the structure and properties of languages and include the following datasets:

- **HellaSwag**: 10,042 multiple choice scenarios in which the model is prompted with a scenario and choose the most likely conclusion to the scenario from four possible options. [37]
- **LAMBADA**: 6,153 passages take from books - we use the formatting adopted by OpenAI's version. [22]
- **Winograd Schema Challenge**: 273 scenarios in which the model must use semantics to correctly resolve the anaphora in a sentence. [17]
- **Winogrande**: 1,267 scenarios in which two possible beginnings of a sentence are presented along with a single ending. [29]

**Symbolic problem solving** tasks test the model's ability to solve a diverse range of symbolic tasks including arithmetic, logical reasoning, algorithms and algebra. These datasets include:

- **BIG-bench algorithms**: 1,320 multiple choice questions. [32]
- **BIG-bench dyck languages**: 1000 complete-the-sequence questions. [32]
- **BIG-bench elementary math QA**: 38,160 four-choice multiple choice arithmetic word problems. [32]
- **BIG-bench operators**: 210 questions involving mathematical operators. [32]
- **GSM8K**: 1,319 short, free-response grade school-level arithmetic word problems with simple numerical solutions. [8]
- **SVAMP**: 300 short, free-response grade school-level arithmetic word problems with simple numerical solutions. [23]

The **Reading comprehension** benchmarks test a model's ability to answer questions based on the information in a passage of text. The datasets include:

- **LSAT-LR**: 510 passage-based four choice multiple choice questions. [38]
- **LSAT-RC**: 268 passage-based four choice multiple choice questions. [38]
- **SAT-English**: 206 passage-based four choice multiple choice questions. [38]
- **CoQA**: 7,983 passage-based short free response questions. [27]

