# OpenReview forum: "Prior-based Noisy Text Data Filtering: Fast and Strong Alternative For Perplexity"
_NeurIPS.cc/2025/Conference — Submitted to NeurIPS 2025_

### Official Review · Reviewer_wNnB · 2025-06-28

**Clarity:** 3
**Significance:** 2
**Originality:** 2
**Rating:** 3
**Confidence:** 4

**Summary:**

This paper proposes a prior-based data filtering method as a fast and powerful alternative to perplexity (PPL)-based filtering. To address the slow speed and unreliability of the PPL approach, this new method is inspired by linguistic frequency analysis and requires no model inference. It works by estimating token "priors" from term frequency and then filtering documents based on the mean and standard deviation of these priors. Experiments show this method is over 1000x faster than PPL-based filtering while achieving superior average performance across 21 downstream benchmarks, and is also effective for symbolic and multilingual corpora

**Questions:**

N/A

**Ethical Concerns:**

["NO or VERY MINOR ethics concerns only"]

**Final Justification:**

Thank you for your effort in organizing the discussion. I'm not satisfied with the author's response. How the model's performance increases along with the number of training tokens is strongly affected by how the evaluation benchmarks are selected. The author's conclusion, "This suggests that performance gaps between baselines are largely established in the early training stages", is observed on the relatively easy benchmarks they selected, while the trend on hard benchmarks like MMLU is still unclear (because MMLU is still near random when the training tokens are limited). This challenges the effectiveness of their method.

In addition, I believe they have made a mistake in common sense by saying "This property is consistent with evidence from the MMLU leaderboard [5] where performance correlates strongly with model size, while smaller models frequently achieve near-random accuracy". Models like Qwen2.5-0.5B/1.5B achieve far better performance than random on MMLU, which means the emergence of capability correlates to the computation, rather than the model sizes.

Therefore, I will not change my score.

**Limitations:**

Yes

**Quality:**

2

**Strengths And Weaknesses:**

Strengths:
1. The proposed method shows a large efficiency improvement over PPL-based methods.
2. The method is easy to implement.

Weakness:
1. The main weakness is the scale of the experiment and evaluation. Since the authors claim that they select data for pre-training, a 3B-tokens training is too small for a valid evaluation of pre-trained models. Besides, the scores on some core benchmarks (like MMLU) are just slightly better than a random guess, making the soundness of the experimental results questionable.
2. The method identifies outliers by measuring deviation from the corpus median of prior-based statistics. While effective for removing noise, this approach risks discarding legitimate, high-quality documents that are simply statistical outliers. For instance, a highly technical paper with rare terminology or a poem with unusual syntax could be incorrectly filtered

---

> ### Author Rebuttal · Authors · 2025-07-31
>
> **[Q] 3B-tokens training is too small for a valid evaluation of pre-trained.**
>
> = We acknowledge the reviewer’s concern. However, we would like to emphasize that our experimental setup follows established practices from recent top-tier ML conference papers [1,2], ensuring rigor despite our resource constraints, considering the nature of pre-training research, which involves multiple long training runs.
>
> In particular, [1] pre-trains on 25M documents with a length of 256, totaling approximately 6B tokens, which is comparable to our setting. Furthermore, [2] demonstrates that performance gaps between models trained on 5B token duration remain consistent across later epochs, and reports that repeating the dataset twice yields the best performance. Following this observation, we repeat our 3B-token dataset twice, resulting in a total training duration of 6B tokens, making our setup comparable to [1] and [2]. As a result, the relative performance of baselines (e.g, DSIR, PPL, no-filter) remained consistent with the findings of prior work.
>
> Therefore, we believe that our evaluation setup is valid and aligned with established prior works, and that it meaningfully extends their experimental foundation.
>
> ---
>
> **[Q] The scores on some core benchmarks (like MMLU) are just slightly better than a random guess.**
>
> =  Notably, significant differences were observed across numerous other representative benchmarks beyond MMLU. Averaging without MMLU would not impact the conclusions of our study.
>
> As you pointed out, the MMLU results are close to random, offering little meaningful information. We included them for only the following minor reasons: (1) our primary reference work [2] adopted the Mosaic Gauntlet with multiple benchmarks, which we followed for consistency; (2) we filtered out benchmarks without meaningful differences by applying a simple criterion: “if the average of baselines was below 0.001”. MMLU marginally met this threshold. (3) While we considered excluding MMLU, we decided to explicitly report that some benchmarks show no significant differences, which we believe is more transparent than omitting them. In any case, near-random performance on MMLU has also been occasionally observed in other LLMs [3].
>
> ---
>
> **[Q]  While effective for removing noise, this approach risks discarding legitimate, high-quality documents that are simply statistical outliers. For instance, a highly technical paper with rare terminology or a poem with unusual syntax could be incorrectly filtered**
>
> = This observation is valid; however, such a cost is an inherent trade-off in all filtering methods, and minimizing this cost is one of the fundamental goals in this research area. Importantly, this trade-off is assessed via benchmark performance, and our approach achieves a better trade-off than previous baselines.
>
> For example, the mentioned problem is observed in symbolic problem-solving benchmarks when using Dolma. Dolma primarily consists of web data ( Common Crawl 74.6%, Table 1), but contains a small amount of code and math scripts (Stack 13.4%), which fits your description of “rare and valuable data.” Indeed, PPL-based models underperform compared to the no-filter baseline on these symbolic tasks, aligning with your observation. Notably, our method achieves higher performance than the PPL-based approach, indicating its superior ability to leverage such high-value minority data.
>
> ** Nevertheless, we introduce two additional methods for a better trade-off. **
>
> (1) Using only stds:
>
> Our original method leverages both the mean and standard deviations (stds) of the prior. Using only the stds may be beneficial, since the mean reflects the average frequency of tokens, whereas the std captures the dynamics among them. This distinction can make stds-based filtering more effective in identifying well-structured documents with low-frequency languages. Empirically, the stds-only filter not only still surpasses baseline approaches in overall performance but also outperforms the stds + mean on symbolic tasks.
>
> (2) Calculate prior on blended corpus:
>
> Our second strategy is to incorporate target-domain data when estimating token frequencies, thereby assigning higher prior probabilities to domain-specific terms and preventing their exclusion during filtering. In practice, we mixed Pile-GitHub data with Dolma in equal proportion for prior computation, which need only additional 10 minutes of processing.
>
> As a result, overall performance improved beyond the original method, while symbolic task performance also increased. (ETable1)
>
> ETable1:  We report the performance of additional variations. The *Prior (Dolma + GitHub)* setting uses both stds and mean. World knowledge and overall scores exclude MMLU, yielding results consistent with the trends reported originally.
>
> |  | Avg | World knowledge | Commonsense reasoning | Language understanding | Symbolic problem solving | Reading comprehension |
> | --- | --- | --- | --- | --- | --- | --- |
> | no-filter | 5.78 | 5.52 | 0.44 | 6.14 | 13.22 | 3.59 |
> | PPL-based | 8.22 | 9.98 | 11.91 | 7.34 | 7.91 | 3.96 |
> | Prior (stds + mean) | 9.20 | 9.53 | 11.27 | 10.31 | 11.13 | 3.79 |
> | Prior  (stds) | 8.70 | 7.28 | 10.57 | 9.34 | 12.40 | 3.89 |
> | Prior (dolma + github) | 9.48 | 11.57 | 10.83 | 8.97 | 12.27 | 3.78 |
>
> ---
>
> ---
>
> **[Q] A highly technical paper with rare terminology or a poem with unusual syntax could be incorrectly filtered.**
>
> = We aim to validate this scenario in a controlled setting.
>
> We first consider a scenario where a document with a general structure contains extremely rare terminology. To simulate this, we sampled 1,000 data points (each  512-token length) from the Dolma dataset within the central ±15% range of $\mu$ and gradually injected rare terms into them. Rare terminology was generated by concatenating two tokens ranked in the bottom 10% of the prior distribution (e.g., “prosecromeda”, combining two tokens “prosec” and “romeda”). We inserted $n$ of these terms into each text and measured the percentage of these texts classified as outliers ( ±25% threshold). The result is as follows (ETable 2)
>
> ETable 2: $n$ is the number of injected terminology, and inliers is the rate of texts remaining in 25%~75% boundary.
>
> | n | n * token length | inliers (25%~75%) |
> | --- | --- | --- |
> | 1 | 2 | 1.0 |
> | 6 | 12 | 1.0 |
> | 7 | 14 | 0.98 |
> | 8 | 16 | 0.91 |
> | 9 | 18 | 0.67 |
>
> Up to seven insertions of rare terms—amounting to 14 tokens (2.7%) within a 512-token block—were required before around 10% of texts were filtered out.
>
>  To assess whether seven occurrences are a reasonable upper bound, we examined the typical frequency of topic-specific terminology in real-world text. Specifically, we sampled 10,000 Wikipedia articles, segmented them into 512-token blocks, normalized all text to lowercase, and counted the occurrences of the article title within each block.
>
> On average, the title appeared 1.09 times per block, corresponding to an average token length of 3.44. This suggests that even documents intended to explain a given concept are far from being dominated by that terminology. Taken together, these findings indicate that the prior-based filter exhibits strong robustness when handling scenarios involving rare terminology.
>
> ---
>
> Reference
>
> [1] S. M. Xie et al., Data selection for language models via importance resampling, NeurIPS 2023
>
> [2] Z. Ankner et al., Perplexed by perplexity: Perplexity-based data pruning with small reference models, ICLR 2025
>
> [3] lmarena-ai/chatbot-arena-leaderboard @ huggingface

---

> > ### Comment · Reviewer_wNnB · 2025-08-04
> >
> > Thank you for your detailed response.
> >
> > I respectfully disagree with the justification for training on only 3B tokens by citing "prior top-tier conference works." My reasoning is as follows:
> >
> > + Relevance of Provided References: The cited works do not adequately support the decision for such a short pre-training schedule in the context of modern LLMs. Reference [1] focuses on training a BERT-like model from scratch, a methodology that is now considerably outdated for contemporary large language model development. Reference [2] does not align with the authors' claim; its final experiments were conducted on much larger data scales (26B and 50B tokens, as shown in its Figure 1), not the smaller scale implied.
> > + Necessity of Longer Pre-training: Shortening the training process based on early-stage metrics is problematic. The pre-training dynamics of LLMs are non-linear and complex. While performance on simpler benchmarks might appear to stabilize early on, this offers no guarantee for how the model will perform on more challenging benchmarks, whose required capabilities may only emerge in later stages of training. The near-random scores on MMLU could be a direct symptom of this issue. A truncated training regimen risks prematurely pruning crucial knowledge from the training corpus, the negative impact of which would only become evident after more extensive training.
> >
> > Therefore, the justification for the 3B token training limit is unconvincing and potentially compromises the model's performance on complex reasoning tasks. Based on this, I am inclined to maintain my original score.

---

> ### Author Response · Authors · 2025-08-04
>
> Thank you for your careful review. However, it seems that part of our reasoning may not have been clearly conveyed.
>
> **#. Regarding reference [2]:**
> As you noted, [2] finally reports performance at 26B and 50B token durations. However, for the 1B model, it also reports results at 5B tokens, and Figure 1 of [2] shows that the performance ranking among baselines remains consistent between 5B and 26B token duration. This suggests that performance gaps between baselines are largely established in the early training stages. Our work leverages this observation.
>
> To elaborate, Section 3.5 of [2] explains that all models are trained with **2× data repetition**. Accordingly, the 5B token duration in Figure 1 corresponds to 2.5B tokens of data repeated twice. Our experimental setup uses 3B tokens repeated twice, which makes it directly comparable to the 5B token point in Figure 1 of [2].
>
> In other words, our experiment effectively reproduces the training regime up to the 5B token mark in [2]. Based on their observation, we have reason to believe that the relative performance ranking among baselines would remain consistent beyond this point.
>
> Additionally, [3] also demonstrates that model performance tends to increase almost linearly with the training FLOPs, while the relative ranking among models remains largely consistent. This suggests that such consistency is a general phenomenon. We attribute this to the fact that most noisy data without meaningful information will remain uninformative regardless of how many times it is trained on. For instance, when examining the noisy data samples we presented fin Figure2, it is unlikely that repeated exposure to these examples would suddenly yield new insights or knowledge.
>
> **#.  Emergence of capability seems to be associated more with model size.**
>
> The sudden emergence of capability you mentioned seems to be more associated with model size than with token duration. This is a well-established property in transformer-based architectures [4].
>
> This property is consistent with evidence from the MMLU leaderboard [5] where performance correlates strongly with model size, while smaller models frequently achieve near-random accuracy. For example, TinyLlama-1.1B, trained on 627B tokens, still performs close to random [6]. Similar patterns can be observed for models such as OPT and Pythia in the 1–1.5B range [6].
>
> From this perspective, the absence of experiments with larger model sizes is a limitation of our paper, while such large-scale pretraining is challenging in most research environments. That said, two observations mitigate this concern: (1) It is reported in [2] that performance trends for 3B models remain consistent with those of 1B models.
> (2) The noise patterns identified in Figure 1 in our paper are still unlikely to suddenly provide meaningful information for significantly larger models.
>
> Furthermore, our method demonstrates strong performance across multiple well-known benchmarks such as HellaSwag and LAMBADA. When applied to datasets like Pile-GitHub, it also achieves SOTA results on symbolic problem solving tasks which is challenging for LLM, indicating its robustness across diverse evaluation settings.
>
> ---
>
> References
>
> [1] S. M. Xie et al., Data selection for language models via importance resampling, NeurIPS 2023
>
> [2]  Z. Ankner et al., Perplexed by perplexity: Perplexity-based data pruning with small reference models, ICLR 2025
>
> [3]  Jeffrey Li et al., DataComp-LM: In search of the next generation of training sets for language models
>
> [4] Alec Radford et al.,  Language Models are Unsupervised Multitask Learners
>
> [5] https://llm-explorer.com/list/?benchmark=hflb_mmlu
>
> [6] Peiyuan Zhang et al., TinyLlama: An Open-Source Small Language Model

---

### Official Review · Reviewer_n2ie · 2025-06-30

**Clarity:** 3
**Significance:** 2
**Originality:** 3
**Rating:** 5
**Confidence:** 3

**Summary:**

This work introduces a novel data selection method based on the mean and std of token priors of a document. The method presents itself as a more computationally efficient and reliable method than its perplexity-based alternative filtering. It analyses the method from various aspects, such as anecdotal evidence of filtered samples as well as showing promising performance in an ablation study with a 1.5B and 137M parameter LLM.

**Questions:**

- How would the authors set their work in the context of recent data curation strategies, beyond perplexity filtering, such as Fineweb-Edu and DCLM?
- In Figure 4) could the authors please clarify the rationale why they consider a 50% overlap as a strong correlation?
- Could the authors please clarify whether they used cloze-style MMLU evaluation? Otherwise, I would be surprised to see MMLU included as a benchmark at this scale (1B model trained on 6B tokens), as I would expect it to perform below random.
- Should appendix section C’s title be: “Details on Benchmarks”?

**Ethical Concerns:**

["NO or VERY MINOR ethics concerns only"]

**Final Justification:**

I thank the authors for their detailed rebuttal. They have addressed my concerns.

**Limitations:**

Yes

**Paper Formatting Concerns:**

No cocern noticed.

**Quality:**

2

**Strengths And Weaknesses:**

**Strengths:**
- This work presents a novel, simple and compute-efficient, but effective alternative to perplexity filtering
- The work is well written and the experimental design is structured and methodical.
- The work analyzes the prior based filter performance from a variety of interesting aspects, such as anecdotal evidence for filtered documents, behavior in multilingual context or overlap with perplexity based filtering.

**Weaknesses**
- The related work is missing key works in data curation (filtering), such as the-state-of-the-art datasets DCLM and Fineweb-Edu and how the contribution of this work is set within that context together with a critical discussion. Potentially, even an experiment with a multilingual dataset such as HPLT or FineWeb-2 would have been interesting to investigate the performance aspect of the language learnability of the prior-based filtering approach.

---

> ### Author Rebuttal · Authors · 2025-07-31
>
> **[Q] The related work is missing key works in data curation (filtering), such as the-state-of-the-art datasets DCLM and Fineweb-Edu.**
>
> = We thank you for pointing out these valuable works, which we will incorporate into the related work section.  However, the concepts discussed in these studies are already well covered by the representative works we reference—[1], [2], and [3]—and thus do not alter the significance of our research.
>
> We will discuss each of the referenced papers, and then the contribution of our work.
>
> (1) DCLM
>
> DCLM introduces benchmarks for evaluating data filtering methods, using Common Crawl as the data pool and assessing performance on two tasks (e.g., MMLU and HellaSwag). The study further explores various filtering strategies (e.g., including heuristic, classifier-based, and PPL-based approaches).
>
> However, both [1] and our work rely on Dolma, which contains approximately 70% Common Crawl (Table 1), along with additional sources such as *The Stack* and PeS20, covering diverse domains and formats. This diversity creates a more challenging setting for filtering, which we believe makes following [1]’s setup experimentally more rigorous. In addition, [1] evaluates language ability across five categories (e.g., knowledge, reasoning, and others) using more than 20 benchmarks, which we consider a more robust evaluation compared to DCLM’s setting.
>
> This work also compares several model-based quality filters, among which fastText, top-k average logits, and PPL-based filtering achieve the highest and comparable performance. Excluding fastText due to its reliance on costly human labeling, PPL-based filtering emerges as the strongest approach—consistent with the findings already reported in [1] and [3].
>
> Thus, the topics addressed in DCLM are already covered in the representative works we primarily reference—[1], [2], and [3]—and we consider these sufficient to encompass the relevant content.
>
> (2) Fineweb-edu
>
> FineWeb-Edu has similar position to DCLM. FineWeb-Edu relies solely on Common Crawl as its data source and evaluates on eight benchmarks, which are also included in our benchmark set.
>
> Most of the methods compared in FineWeb-Edu are based on heuristic filtering, which includes URL-based exclusions and deduplication. However, prior studies have noted that model-based approaches are more advanced in performance [2, 4], which positions our work as tackling a more advanced filtering paradigm. Moreover, rule-based methods are entirely compatible with our approach, as those techniques are already commonly employed at the web crawling level [4].
>
> *. Contribution of our work within those related works.
>
> As summarized, the primary works we reference [1, 2, 3] encompass many concepts and baselines explored in related literature and identify PPL-based filtering and DSIR as state-of-the-art methods offering strong performance at a reasonable cost. Our contribution lies in proposing an approach that achieves even better performance while operating at an extremely lower computational cost.
>
> ---
>
> **[Q] An experiment with a multilingual dataset such as HPLT or FineWeb-2 would have been interesting.**
>
> = Thank you for suggesting these valuable datasets; we will consider incorporating them into the camera-ready version if time permits.
>
> However, we note that Dolma inherently embodies the complexity of a multilingual setting, given its composition: a predominant share of English text from Common Crawl (74.6%) alongside a smaller but significant fraction of other language types, such as programming code and mathematical content from The Stack (13.4%).  And both linguistic ability (general domain capability and symbolic task ability) are separately evaluated.
>
> Empirical findings on Dolma reveal issues likely to arise in multilingual setting. While the PPL-based model are improved in general linguistic ability, its performance on symbolic language tasks falls below that of the no-filter baseline. This pattern suggests that the model is biased toward the dominant language, resulting in the misclassification of minority language types as noise.
>
> Our prior-based filter outperforms PPL on symbolic tasks, indicating better coverage across both language types in mixed settings. Nevertheless, the prior filter still underperforms the no-filter baseline on symbolic tasks, leaving room for improvement. To address this, we propose two additional refinements.
>
> **Additional methods for better coverage of mixed languages**
>
> (1) Using only stds:
>
> Our original method leverages both the mean and standard deviations (stds) of the prior. Using only the stds may be beneficial, since the mean reflects the average frequency of tokens, whereas the std captures the dynamics among them. This distinction can make stds-based filtering more effective in identifying well-structured documents with low-frequency languages. Empirically, the stds-only filter not only still surpasses baseline approaches in overall performance but also outperforms the stds + mean on symbolic tasks.
>
> (2) Calculate prior on blended corpus:
>
> Our second strategy is to incorporate target-domain data when estimating token frequencies, thereby assigning higher prior probabilities to domain-specific terms and preventing their exclusion during filtering. In practice, we mixed Pile-GitHub data with Dolma in equal proportion for prior computation, which need only additional 10 minutes of processing.
>
> As a result, overall performance improved beyond the original method, while symbolic task performance also increased.
>
> ETable1:  We report the performance of additional variations. The *Prior (Dolma + GitHub)* setting uses both stds and mean. World knowledge and overall scores exclude MMLU, yielding results consistent with the trends reported originally.
>
> |  | Avg | World knowledge | Commonsense reasoning | Language understanding | Symbolic problem solving | Reading comprehension |
> | --- | --- | --- | --- | --- | --- | --- |
> | no-filter | 5.78 | 5.52 | 0.44 | 6.14 | 13.22 | 3.59 |
> | PPL-based | 8.22 | 9.98 | 11.91 | 7.34 | 7.91 | 3.96 |
> | Prior (stds + mean) | 9.20 | 9.53 | 11.27 | 10.31 | 11.13 | 3.79 |
> | Prior  (stds) | 8.70 | 7.28 | 10.57 | 9.34 | 12.40 | 3.89 |
> | Prior (dolma + github) | 9.48 | 11.57 | 10.83 | 8.97 | 12.27 | 3.78 |
>
> ---
>
> **[Q] In Figure 4, could the authors please clarify the rationale why they consider a 50% overlap as a strong correlation?**
>
> = The degree of correlation can be discussed in relative terms. Our method (prior mean and stds) achieves an overlap substantially higher than random. For example, with e = 10, random sampling yields an expected overlap of 10%, while the μ-based filter achieves significantly higher overlap. Of course, the overlap is lower than 100%. This is likely because prior value captures different properties as described (Figure 3), and because PPL itself is unstable. (Line 184).
>
> **Additional analysis**
>
> We present an additional analysis illustrating the correlation between the prior and PPL. Our hypothesis is that this correlation would be more pronounced among outliers, as inlier data exhibit very low variance, making rankings fluctuate easily.
>
> To examine this, we selected the top, middle, and bottom 1,000 samples based on μ and computed their average PPL (E(PPL), see ETable 2). While the results initially appear to show a highly linear correlation, this is largely driven by extreme outliers within each subset. To mitigate this effect, we recalculated the average after trimming the top and bottom five samples from each subset ( E(PPL*) ), and then measured the 1-norm difference from the median PPL: |E(PPL*) - M(PPL)|.
>
> The results reveal that subsets centered on μ correspond to median PPL values, whereas subsets with extreme μ values (top and bottom) exhibit PPL averages that are substantially distant from the median. This indicates that prior-based rankings effectively identify segments of the data that deviate from the central distribution under PPL.
>
> ETable2. 1000 samples from the top, middle, and bottom ranks.
>
> |  | E(PPL) | E(PPL*) trimmed | abs(E(PPL*) - M(PPL)) |
> | --- | --- | --- | --- |
> | top μ | 1.5 | 1.4 | 33.1 |
> | middle  μ|460.2 | 34.7 | 0.1 |
> | bottom μ| 4701.9 | 9.0 | 25.6 |
>
> ---
>
> **[Q] Metric used with MMLU. \\  MMLU exhibits no significant difference.**
>
> All evaluation settings follow the Gauntlet [5] protocol. For MMLU, the model computes token-wise perplexity for each choice and selects the option with the lowest perplexity as the predicted answer.
>
> As you pointed out, the MMLU results are close to random (Table 4), offering little meaningful information. We included them for only the following minor reasons: (1) our primary reference work [2] adopted the Mosaic Gauntlet with multiple benchmarks, which we followed for consistency; (2) we filtered out benchmarks without meaningful differences by applying a simple criterion: “if the average of baselines was below 0.001”. MMLU marginally met this threshold. (3) While we considered excluding MMLU, we decided to explicitly report that some benchmarks show no significant differences, which we believe is more transparent than omitting them.  In any case, near-random performance on MMLU has also been occasionally observed in other LLMs [6].
>
> Notably, significant differences were observed across numerous other representative benchmarks beyond MMLU. And average performance excluding MMLU exhibit consistent trends (ETable 1).
>
> ---
>
> [1] Z. Ankner et al., Perplexity-based data pruning with small reference models, ICLR 2025
>
> [2] S. M. Xie et al., Data selection for language models via importance resampling, NIPS 2023
>
> [3] M. Marion et al., When less is more: Investigating data pruning for pretraining llms at scale
>
> [4] Jeffrey Li et al., DataComp-LM: In search of the next generation of training sets for language models,  NeurIPS 2025 Dataset and Benchmark Track
>
> [5] https://github.com/mosaicml/llm-foundry.git
>
> [6] lmarena-ai/chatbot-arena-leaderboard @ huggingface

---

> > ### Comment · Reviewer_n2ie · 2025-08-04
> >
> > I thank the authors for their detailed rebuttal. I would like to note that DCLM and FineWeb-Edu represent current state-of-the-art Common Crawl open-source datasets, both demonstrating superior performance compared to Dolma in their respective benchmarks. Given that combining high-performing datasets from different resources is standard practice in the field, I would encourage the authors to consider incorporating DCLM and FineWeb-Edu as relevant work beyond [1, 2, 3].
> >
> > I would also appreciate clarification on the following points:
> > 1. "Most of the methods compared in FineWeb-Edu are based on heuristic filtering, which includes URL-based exclusions and deduplication." - My understanding is that FineWeb-Edu was created using a model-based filtering approach rather than heuristic methods.
> > 2. "Excluding fastText due to its reliance on costly human labeling" - The DCLM dataset was filtered using a fastText model that utilized OH2.5 and Reddit data samples as positives, without requiring human labels.
> >
> > I thank the authors for further clarification.

---

> > > ### Author Response · Authors · 2025-08-05
> > >
> > > We acknowledge the importance of DCLM and FineWeb-Edu and will include a discussion of these works in the related work section. However, we would like to provide the following clarification:
> > >
> > > ---
> > >
> > > **[Q1-1] Given that combining high-performing datasets from different resources is standard practice in the field, I would encourage the authors to consider incorporating DCLM and FineWeb-Edu as relevant work beyond**
> > >
> > > = In terms of data sources, both DCLM and FineWeb rely exclusively on Common Crawl (CC). CC is also included in Dolma, which additionally incorporates multiple other datasets. Given that “combining datasets from different resources is standard practice,” Dolma not only includes the sources of DCLM and FineWeb but also extends them with broader diversity.
> > >
> > > In this reason, Dolma provides a more suitable test environment for our research focus, as it explicitly contains a small proportion of symbolic language data (e.g., *The Stack*), introducing a multi language type and multi-domain challenge.
> > >
> > > Our work does not aim to build or release the best dataset but rather to investigate filtering methods. Under this perspective, Dolma serves as a more rigorous and representative evaluation setting.
> > >
> > > **[Q1-2] DCLM and FineWeb-Edu represent current state-of-the-art Common Crawl open-source datasets, both demonstrating superior performance compared to Dolma in their respective benchmarks.**
> > >
> > > =  DCLM uses data sources similar to those in Dolma. However, its superior performance primarily stems from the application of more extensive data filtering. Since our research assumes applying a filtering method to the dataset and evaluating, a direct performance comparison between DCLM and Dolma is not meaningful. In fact, for our purposes, we require data that is as close as possible to the original, minimally filtered source, which makes Dolma a more suitable choice.
> > >
> > > ---
> > >
> > > **[Q2] "Most of the methods compared in FineWeb-Edu are based on heuristic filtering, which includes URL-based exclusions and deduplication." - My understanding is that FineWeb-Edu was created using a model-based filtering approach rather than heuristic methods.**
> > >
> > > = We understood—FineWeb primarily relies on heuristic-based filtering, while FineWeb-Edu adopts a model-based approach. However, FineWeb-Edu use huge model (LLaMA-3-70B) to label 460k samples, and employs an embedding model for inference-based classification. This process is extremely resource- and time-intensive, placing it outside the scope of our work.
> > >
> > > ---
> > >
> > > **[Q3] "Excluding fastText due to its reliance on costly human labeling" - The DCLM dataset was filtered using a fastText model that utilized OH2.5 and Reddit data samples as positives, without requiring human labels.**
> > >
> > > = We understand that positive and negative data were manually curated. However, this approach raises several concerns:  (1) As shown in Table 4 of [1], FastText exhibits similar performance with the perplexity-based filtering, 0.4 point difference on the Extended set. (2) The effectiveness of FastText is likely sensitive to the manual curation. In fact, FastText is also compared with DSIR in [2], and reported lower performance than DSIR. (3) Manual curation introduces inherent constraints, particularly in multilingual or multi-domain settings. For example, Dolma blends multiple language types and symbolic data, making it impractical to curate data for each type. As described in Section 3.4.2, in a multilingual setting, FastText would require manually defining positive and negative datasets for each language. This introduces substantial complexity.
> > >
> > > This might explain why FastText performs reasonably well in DCLM, which relies solely on Common Crawl, but would face challenges in more heterogeneous datasets like Dolma.
> > >
> > > ---
> > >
> > > [1] Jeffrey Li et al., DataComp-LM: In search of the next generation of training sets for language models
> > >
> > > [2]  S. M. Xie et al., Data selection for language models via importance resampling

---

> ### Comment · Area_Chair_hs2v · 2025-08-04
> **Reviewer n2ie: please respond to the rebuttal of Submission17521**
>
> Hi Reviewer n2ie,
>
> Please respond to the authors' rebuttal ASAP to see if the authors have addressed your concerns.
>
> Sincerely,
>
> Your AC

---

> ### Author Response · Authors · 2025-08-07
> **Additional analysis**
>
> We additionally conducted experiments with **FastText**, using the model and code provided by DCLM paper [1]. The results were consistent with [1]:  FastText only slightly outperformed perplexity-based filtering, but **the prior-based method clearly outperformed (ETable 1).** This validates the robustness of our approach.
>
> $\color{white}$
>
> ETable 1. Performance of small (137M) models on Dolma.
>
> |  | Avg | World knowledge | Commonsense reasoning | Language understanding | Symbolic problem solving | Reading comprehension |
> | --- | --- | --- | --- | --- | --- | --- |
> | no-filter | 4.68 | 3.59 | 1.81 | 1.47 | 12.83 | 3.70 |
> | PPL-based | 4.92 | 3.75 | 6.53 | 2.90 | 7.84 | 3.58 |
> | FastText | 4.94 | 3.36 | 0.52 | 3.04 | 13.85 | 3.92 |
> | Prior-based | 6.26 | 3.10 | 9.13 | 4.22 | 11.21 | 3.66 |
>
> To clarify, the DCLM's FastText model was trained to predict the **probability** that a given sample belongs to a high-quality set. According to [1], web-crawled data (RefinedWeb[2], which is based on Common Crawl) was assigned the “low-quality” label, while ELI5 and OH-2.5 were assigned the “high-quality” label. During data selection, we retained only the top 50% of samples ranked by their predicted probability of belonging to the high-quality set, ensuring that size of training dataset matched that of other baselines.
>
> **Case analysis**
>
> We analyzed the outlier cases from the FastText classifier and observed several unexpected patterns. Among the samples classified as having a 0% probability of belonging to the high-quality set, we observed a substantial number of well-formed texts—particularly those resembling news articles (**Case 1**). Conversely, many samples classified as having a 100% probability of being high-quality were clearly noisy or nonsensical scripts (**Case 2**). We double-checked our implementation, but confirmed that these results were solely due to the model’s inference behavior.
>
> $\color{white}$
>
> Case1.  Data assigned a 0% probability of being high-quality (i.e., 100% belonging to RefinedWeb) — ranking at the absolute bottom of the distribution.
>
> ```
> Paula’s Choice – Donating $50,000 to the COVID-19 Solidarity Response Fund for World Health Organization.\nPyer Moss – Pyer Moss has set aside $10,000 to get supplies for medical workers while also converting their NYC office into a donation center to store the supplies. Using local factories, Pyer Moss is creating 1,000 mask covers to send directly to front line workers. With the
> ```
>
> $\color{white}$
>
> Case2. Data assigned a 100% probability of being high-quality — ranking at the absolute top of the distribution.
>
> ```
> '0][1-9]\\|[1][0-2])([0-2][0-9]\\|[3][0-1])\\\\\\\\s\\\\\\\\s?([0-1]?[0-9]\\|[2][0-3]):[0-5][0-9]:[0-5][0-9])` | | |\n| `ddMMyy HH:mm:ss` | `(([0-2][0-9]\\|[3][0-1])([0][1-9]\\|[1][0-2])[0-9]{2}\\\\\\\\s\\\\\\\\s?([0-1]?[0-9]\\|[2][0-3]):[0-5][0-9]:[0-5][0-9])` | | |\n| `MMM d HH:mm:ss` | `(Jan\\|Feb\\|Mar\\|Apr\\|May\\|Jun\\|Jul\\|Aug\\|Sep\\|Oct\\|Nov\\|Dec)\\\\\\\\s\\\\\\s
> ```
>
> $\color{white}$
>
> We hypothesize the following reasons for this behavior, which may also explain the suboptimal performance of the FastText filter:
>
> **(1) RefinedWeb contains a substantial amount of well formed data.**
>
> To avoid the cost of human labeling, DCLM chose to label RefinedWeb as the low-quality set, while labeling OH-2.5 and ELI5 as the high-quality set. During inference, the model predicts the probability (0~100%) of input text to belong high-quality set. However, RefinedWeb also contain considerable proportion of well-formed documents. As a result, many samples with 0% probability are in fact well-formed and informative texts (Case1), often resembling those found in RefinedWeb.
>
> One possible explanation is that news article–style texts are prevalent in web-crawled sources (e.g., RefinedWeb), but largely absent from curated datasets like OH-2.5 and ELI5, which mainly contain question-answering format. As a result, the FastText model may have implicitly learned to classify the news article (or other non-QA-style) format as belonging to low-quality set, leading to systematic misclassification.
>
> **(2) Limited discrimination capacity.**
>
> Another possible explanation is the prevalence of code and math-related content in OH-2.5. Since such sources are relatively less common in RefinedWeb, the model may have overfit to these symbolic patterns during training. However, due to the limited capacity of the small FastText model, it is unable to capture deeper coherence within symbolic language. As a result, it may incorrectly classify meaningless noise that superficially resembles symbolic content (as in **Case 2**) as belonging to the high-quality set with 100% probability.
>
> These results underscore the weaknesses of model-based methods, supporting the robustness of prior-based approach.
>
> ---
>
> [1] Jeffrey Li et al., DataComp-LM: In search of the next generation of training sets for language models
>
> [2] Guilherme Penedo et al., The RefinedWeb dataset for Falcon LLM

---

> > ### Comment · Reviewer_n2ie · 2025-08-07
> >
> > I thank the authors for their detailed feedback. I raise my score to a positive one. I still encourage the authors to expand the related work section to better contextualize their contributions.

---

> > > ### Author Response · Authors · 2025-08-08
> > >
> > > We sincerely appreciate your careful consideration of our response, as well as the score increase. We will ensure that your suggestions are carefully reflected in the camera-ready version.

---

### Official Review · Reviewer_ybRV · 2025-07-02

**Clarity:** 4
**Significance:** 4
**Originality:** 4
**Rating:** 6
**Confidence:** 4

**Summary:**

The paper introduces a method to filter data sources for language model training using term-frequency based priors. They contrast it with previous perplexity-based methods, and demonstrate that their method is equally effective while being significantly cheaper and faster. They also show that the method can be applied in multilingual settings, as well as specialized languages like code.

**Questions:**

1. When using Dolma, did you include The Stack ? There’s work that shows that having some “noise” in your pretraining data is helpful overall. In this case, is it then better to apply the prior-based method to each data source individually?

**Ethical Concerns:**

["NO or VERY MINOR ethics concerns only"]

**Final Justification:**

No changes to my rating. The method presented in the paper is simple, and has practical uses for data source filtering. My one comment on minority data sources has also been addressed.

**Limitations:**

Mentioned above.

**Paper Formatting Concerns:**

Lines 44-53 and 96-104 have repeated sentences. This can be rewritten slightly.

**Quality:**

4

**Strengths And Weaknesses:**

Strengths:

1. Very simple and elegant solution, utilizing linguistics.
2. The analysis is well-organized and thorough.
3. The method is not only faster and cheaper compared to PPL, it’s also useful in cases when a reference model might be too small to learn the prior well. The prior-selected documents also overlap heavily with the PPL-selected ones.
4. The prior-based method does not require careful curation of reference dataset.

Weaknesses:

1. This is not a weakness of only this method per se, but all automated filtering methods - they only include what is most common. For certain cases, or to ensure that even minority data sources are represented in the data, you may still require heuristics-based filtering. It will be helpful to discuss this in the limitations.

---

> ### Author Rebuttal · Authors · 2025-07-31
>
> Thank you for your positive assessment of our method. We also appreciate your concern and provide the following discussion.
>
> ---
>
> **[Q] Difficulty in preserving minority data**
>
> = As you correctly noted, this is an inherent cost paid by all filtering methods, and reducing this cost remains one of the goals in this field. We will include this point in the limitations section of the paper.
>
> Nevertheless, we make the following observations and propose improved approaches to address this issue.
> First, the aspect you mentioned has been partially measured through benchmarks, where our baselines demonstrate relative strengths.  For example, the scenario you described can be observed in the performance on symbolic task benchmarks within Dolma. Dolma primarily consists of English-based web data (Common Crawl, 74.6%, Table 1) but includes a small fraction of code and mathematical content (Stack, 13.4%), which can be considered the type of “high-value data with minor language” you referenced.
>
> Notably, PPL-based model report decreased performance on these symbolic benchmarks compared to no-filter, trading off symbolic reasoning ability for improved general reasoning performance, which matches to your concern. By contrast, our method achieves higher performance on symbolic tasks than PPL, indicating that it is more effective at retaining and leveraging such minority data.
>
> Nevertheless, as prior-based method also exhibit relatively lower performance in symbolic ability compared to no-filter, we suggest two more improvement to minimize this trade-off.
>
> (1) Using only stds:
>
> Our original method leverages both the mean and standard deviations (stds) of the prior. Using only the stds may be beneficial, since the mean reflects the average frequency of tokens, whereas the std captures the dynamics among them. This distinction can make stds-based filtering more effective in identifying well-structured documents with low-frequency languages. Empirically, the stds-only filter not only still surpasses baseline approaches in overall performance but also outperforms the stds + mean on symbolic tasks.
>
> (2) Calculate prior on blended corpus:
>
> Our second strategy is to incorporate target-domain data when estimating token frequencies, thereby assigning higher prior probabilities to domain-specific terms and preventing their exclusion during filtering. In practice, we mixed Pile-GitHub data with Dolma in equal proportion for prior computation, which need only additional 10 minutes of processing.
>
> As a result, overall performance improved beyond the original method, while symbolic task performance also increased.
>
> ETable 1:    We report the performance of additional variations. The *Prior (Dolma + GitHub)* setting uses both stds and mean. World knowledge and overall scores exclude MMLU.
>
> |  | Avg | World knowledge | Commonsense reasoning | Language understanding | Symbolic problem solving | Reading comprehension |
> | --- | --- | --- | --- | --- | --- | --- |
> | no-filter | 5.78 | 5.52 | 0.44 | 6.14 | 13.22 | 3.59 |
> | PPL-based | 8.22 | 9.98 | 11.91 | 7.34 | 7.91 | 3.96 |
> | Prior (stds + mean) | 9.20 | 9.53 | 11.27 | 10.31 | 11.13 | 3.79 |
> | Prior  (stds) | 8.70 | 7.28 | 10.57 | 9.34 | 12.40 | 3.89 |
> | Prior (dolma + github) | 9.48 | 11.57 | 10.83 | 8.97 | 12.27 | 3.78 |
>
> ---
>
> **[Q] There’s work that shows that having some “noise” in your pretraining data is helpful overall.**
>
> = Thank you for introducing such an interesting paper. We believe the work you mentioned is likely *“Do We Really Have to Filter Out Random Noise in Pre-Training Data for Language Models?”*  This is indeed a fascinating study and, represents one of the promising future directions.
>
> That said, we would like to share several perspectives on this work
>
> (1) controlability: Regardless of whether noise is beneficial, having greater control is always advantageous, as it enables systematic investigation of the optimal type and amount of noise to include.
>
> (2) resource efficiency: Pre-training entails massive training time and resources. Filtering is intended not only to improve model performance but also to enhance efficiency. By removing noisy data, the model can be exposed to the same amount of informative content with fewer training steps.
>
> (3) evaluation from other work:  The referenced paper reports that models trained with different noise proportion exhibit almost similar downstream task performance, within the error bar. However, this finding is based on only four text classification benchmarks.
>
> In contrast, [1] demonstrates a clear advantage of applying PPL-based filtering when models are trained for sufficient epochs and evaluated across a much broader set of benchmarks. Notably, [1] also observes that this performance gap tends to widen as training progresses.
>
> Reconciling these two claims and rigorously determining their validity remains an important direction for future research.
>
> ---
>
> References
>
> [1] Z. Ankner et al., Perplexed by perplexity: Perplexity-based data pruning with small reference models, ICLR 2025

---

### Official Review · Reviewer_iMnf · 2025-07-11

**Clarity:** 4
**Significance:** 2
**Originality:** 2
**Rating:** 5
**Confidence:** 4

**Summary:**

The paper proposes a simple yet effective filtering method, called prior-based filtering, which identifies outlier documents based on corpus-level token frequency statistics—specifically the mean and variance of token priors.

**Questions:**

1. Have you compared prior-based filtering against other lightweight heuristic filters—such as sentence length thresholds, or script-based rules? Given the shared goal of efficiency, it would be helpful to understand whether prior-based filtering meaningfully outperforms these simpler baselines.

2. Your method offers a fully unsupervised and model-free alternative to PPL-based filtering, which is a key strength. That said, many practical pipelines use lightweight supervised classifiers like fastText or simple quality classification. While fastText requires labeled data and may not be directly comparable, have you considered benchmarking against such classifier-based filters as a complementary baseline? It would help clarify the unique strengths of prior-based filtering relative to real-world, efficient alternatives.

3. In multilingual corpora, your method detects the transition from unlearnable (noise) to learnable language clusters via mean priot. But in truly low-resource settings—where valuable language data never reaches that threshold—does this method risk filtering it out entirely? How might this be mitigated for underrepresented languages?

4. Your method relies on token frequency statistics, which are tokenizer-dependent. Have you evaluated how sensitive the filtering results and downstream performance are to the choice of tokenizer (e.g., GPT-2 vs. multilingual or domain-specific tokenizers)?

5. Your method operates at the document level, using 512-token blocks to compute the mean and standard deviation of token priors. Have you explored how sensitive the filtering behavior and downstream performance are to the choice of block size? For instance, would larger or smaller blocks change the robustness of the statistics or the ability to detect noisy samples?

**Ethical Concerns:**

["NO or VERY MINOR ethics concerns only"]

**Final Justification:**

I would like to thank the authors for their detailed and thoughtful response, as well as for conducting additional experiments to address my questions. I believe they have sufficiently addressed my concern, and I would increase my score to Accept.

As a suggestion for strengthening the final submission, it may be valuable to include a discussion on handling low-resource languages—particularly comparing the std-only, mean-only, and combined std+mean approaches—along with the ablation study on the tokenizer.

**Limitations:**

Good enough

**Quality:**

3

**Strengths And Weaknesses:**

**Strengths:**

(+) Extremely fast and scalable compared to PPL-based filtering methods

(+) Model-free and fully unsupervised, requiring no labeled data or reference corpus

(+) Conceptually simple yet grounded in linguistic intuition; the paper is clearly written and easy to follow

**Weaknesses:**

(−) No comparison against other efficient, lightweight filtering baselines (e.g., heuristic filters or fastText classifiers)

(−) Lacks in-depth analysis of sensitivity to design choices such as block size, filtering thresholds, and tokenizer granularity

---

> ### Author Rebuttal · Authors · 2025-07-31
>
> **[Q] There is a lack of comparison with other lightweight heuristic and classifier-based filters.**
>
> = We acknowledge this concern. However, due to the nature of our study, which requires a long period of pre-training of multiple baselines, we faced significant resource and time constraints. To ensure experimental rigor under these limitations, we delicately adopted settings from recent representative works published in top-tier conferences such as NeurIPS and ICLR within the past 1–2 years ([1], [2]), and representative works such as [3], serving as primary references. [5] is also a meaningful reference. Grounded in these references, we provide a detailed discussion of each category.
>
> (1)  Classifier-based (model-based) method
>
> Previous studies [1, 2, 3, 5] have demonstrated that various model-based filters perform worse than our main baselines, DSIR and PPL-based. For example, [1] shows that PPL-based filtering outperforms classifier models trained on manually curated data. Similarly, [3] reports that PPL-based methods clearly surpass approaches leveraging error or memorization scores derived from model predictions. Furthermore, [5] finds that PPL-based filtering significantly outperforms linear classifiers trained on pretrained BPE text embeddings.
>
> Taken together, these findings establish DSIR and PPL-based filtering as the most effective model-based methods to date. Consequently, demonstrating that our method surpasses these baselines provides strong evidence of its effectiveness over other model-based methods.
>
> (2) Heuristic method
>
> We agree that lightweight heuristic filtering can be useful. However, two important points should be emphasized. 1) Similar to model-based approaches, heuristic filters have been empirically shown to underperform compared to our primary baseline, DSIR, as reported in [1]. Likewise, [5] demonstrates that PPL-based methods significantly outperform a wide range of rule-based heuristic filters. Since our method outperforms both DSIR and PPL, we consider this as providing indirect yet strong evidence of its superiority over heuristic approaches.
>
>    2). As heuristic filters are extremely cost-efficient, they are also highly compatible with our method in practice. They are usually already applied at web crawling level, even including Dolma [4]. The fact that our approach still delivers additional performance gains with Dolma demonstrates its compatibility with heuristic filter.
>
> ---
>
> **[Q] In  valuable language data never reaches the threshold, does this method risk filtering it out entirely?**
>
> = Such a risk is an inherent trade-off in all filtering methods, and minimizing this cost is one of the goals in this area. Importantly, this trade-off is assessed via benchmark performance, and our approach achieves a better trade-off than previous baselines.
>
> For example, the mentioned problem is observed in symbolic problem-solving benchmarks when using Dolma. Dolma primarily consists of web data ( Common Crawl 74.6%, Table 1), but contains a small amount of code and math scripts (Stack 13.4%), which fits your description of “rare and valuable data.” Indeed, PPL-based models underperform compared to the no-filter baseline on these symbolic tasks, aligning with your observation. Notably, our method achieves higher performance than the PPL-based approach, indicating its superior ability to leverage such high-value minority data.
>
> ** Nevertheless, we introduce two additional methods for a better trade-off. **
>
> (1) Using only stds:
>
> Our original method leverages both the mean and standard deviations (stds) of the prior. Using only the stds may be beneficial, since the mean reflects the average frequency of tokens, whereas the std captures the dynamics among them. This distinction can make stds-based filtering more effective in identifying well-structured documents with low-frequency languages. Empirically, the stds-only filter not only still surpasses baseline approaches in overall performance but also outperforms the stds + mean on symbolic tasks.
>
> (2) Calculate prior on blended corpus:
>
> Our second strategy is to incorporate target-domain data when estimating token frequencies, thereby assigning higher prior probabilities to domain-specific terms and preventing their exclusion during filtering. In practice, we mixed Pile-GitHub data with Dolma in equal proportion for prior computation, which need only additional 10 minutes of processing.
>
> As a result, overall performance improved beyond the original method, while symbolic task performance also increased.
>
> ETable 1:  We report the performance of additional variations. The *Prior (Dolma + GitHub)* setting uses both stds and mean. World knowledge and overall scores exclude MMLU.
>
> |  | Avg | World knowledge | Commonsense reasoning | Language understanding | Symbolic problem solving | Reading comprehension |
> | --- | --- | --- | --- | --- | --- | --- |
> | no-filter | 5.78 | 5.52 | 0.44 | 6.14 | 13.22 | 3.59 |
> | PPL-based | 8.22 | 9.98 | 11.91 | 7.34 | 7.91 | 3.96 |
> | Prior (stds + mean) | 9.20 | 9.53 | 11.27 | 10.31 | 11.13 | 3.79 |
> | Prior  (stds) | 8.70 | 7.28 | 10.57 | 9.34 | 12.40 | 3.89 |
> | Prior (dolma + github) | 9.48 | 11.57 | 10.83 | 8.97 | 12.27 | 3.78 |
>
> ---
>
> **[Q] Sensitivity to block size**
>
> We conduct the following analysis to assess whether outliers remain consistently detected across different block sizes.
>
> We randomly concatenated two 512-token blocks to make 1024-token blocks, then trimmed $e/2\%$ from each side to identify outlier samples: $x^{1024}i = x^{512}{2i} \oplus x^{512}_{2i+1},$
>
> where i denotes the data index in the corpus X, and $x^n$ represents a text block of size n, with $x^{512} \in X^{512}$.
>
> If a sample $x_i^{512}$ is classified as a $e\%$ outlier, we then check whether the concatenated block $x^{1024}_{i//2}$ is also classified as an outlier. We repeat the same comparison for $x^{2048}$.
>
> As shown in ETable 2, outliers of smaller blocks were largely retained as outliers in larger blocks, indicating strong alignment. However, the overlap diminishes as n grows. This is because if a page contains both noisy and (more or equal proportion of) clean content, discarding the entire page may not always be ideal.
>
> ETable 2. Overlaps of outliers between dataset with different block size.
>
> |  | n=1024 | n=2048 |
> | --- | --- | --- |
> | e=5 | 0.7935 | 0.6954 |
> | e=10 | 0.8145 | 0.7263 |
> | e=20 | 0.8102 | 0.7265 |
>
> ---
>
> **[Q] Sensitivity to tokenizer**
>
> We additionally compare two tokenizers: LLaMA-3-8B (UTF-8-based, vocab size 128K), and T5-small (SentencePiece based, vocab size 32K). GPT-2 tokenizer in our paper has vocab size 50K. Following the previous experiment, we examine the overlap of outliers (top and bottom $e/2\%$).
>
> ETable 3. Overlaps of outliers between the tokenizers of each model and GPT-2.
>
> | e  | LLaMA-3-8B | T5-small |
> | --- | --- | --- |
> | 0.01 | 0.7734 | 0.8978 |
> | 0.1 | 0.7558 | 0.8709 |
> | 0.2 | 0.7332 | 0.8607 |
> | 0.5 | 0.7349 | 0.8588 |
>
> The outliers appear to be largely consistent across tokenizers, with the T5 exhibiting stronger alignment than LLaMA-3. We further conducted an experiment to determine which tokenizer yields better performance. Due to resource constraints, this evaluation was performed using only a small model.
>
> ETable 4. Performance of small (137M) models.
>
> |  | Avg | World knowledge | Commonsense reasoning | Language understanding | Symbolic problem solving | Reading comprehension |
> | --- | --- | --- | --- | --- | --- | --- |
> | no-filter | 4.68 | 3.59 | 1.81 | 1.47 | 12.83 | 3.70 |
> | PPL-based | 4.92 | 3.75 | 6.53 | 2.90 | 7.84 | 3.58 |
> | Prior | 6.26 | 3.10 | 9.13 | 4.22 | 11.21 | 3.66 |
> | Prior  (LLaMA3) | 5.64 | 3.62 | 5.55 | 4.27 | 10.70 | 4.07 |
> | Prior (T5) | 5.35 | 2.59 | 4.70 | 4.14 | 11.66 | 3.66 |
>
> The results show that all tokenizers outperform the main baselines, regardless of tokenizer choice. However, GPT-2 tokenizer is still the best. Therefore, there was no linear correlation between performance and vocab size. We propose three possible interpretations: (1) Retokenizing the original data with alternative tokenizers introduced variance in sequence lengths, which may have added noise to the prior’s mean and stds distributions, reducing performance. (2) Using the tokenizer paired with the model may offer stability. (3) There may exist an optimal granularity, which is that of GPT-2.
>
> Nevertheless, as demonstrated, priors can be computed even with tokenizers mismatched to the model. If an optimal tokenizer exists, it can be freely adopted any time.
>
> ---
>
> **[Q] Sensitivity to threshold**
>
> As with other aspects of our work, resource constraints inherent to pre-training research necessitated leveraging prior findings. In particular, our main reference [2] extensively evaluated PPL-based filtering across different selection rates (25%, 50%, 75%) and concluded that 50% was the most effective. Since our method is conceptually intended to serve as an alternative to PPL-based filtering, we adopted this optimal configuration (50%) for fair comparison. Consequently, our approach can be regarded as a viable substitute for the most effective PPL-based setting.
>
> Nevertheless, we explored an additional approach for verification. As illustrated in Figure 2, we normalized $\mu$, $\sigma$, and rank to a 0~1 scale and defined boundaries where the gradient crosses -1 (on both sides), resulting in an 82% threshold of the data. However, the overall score was 0.08169 (Large model), lower than the 50% threshold.
>
> ---
>
> [1] S. M. Xie et al., Data selection for language models via importance resampling, NIPS 2023
>
> [2]Z. Ankner et al., Perplexity-based data pruning with small reference models, ICLR 2025
>
> [3] M. Marion et al., Investigating data pruning for pretraining LLMs at scale
>
> [4] Luca Soldaini et al., An Open Corpus of Three Trillion Tokens for Language Model Pretraining Research
>
> [5] Jeffrey Li et al., In search of the next generation of training sets for language models

---

> > ### Comment · Reviewer_iMnf · 2025-08-04
> >
> > I would like to thank the authors for their detailed and thoughtful response, as well as for conducting additional experiments to address my questions. I believe they have sufficiently addressed my concerns, and I would increase my score to Accept.
> >
> > As a suggestion for strengthening the final submission, it may be valuable to include a discussion on handling low-resource languages—particularly comparing the std-only, mean-only, and combined std+mean approaches—along with the ablation study on the tokenizer.

---

> > > ### Author Response · Authors · 2025-08-08
> > >
> > > We sincerely appreciate your thoughtful engagement with our response and for raising the score. We will make sure to carefully incorporate all of your suggestions into the camera-ready version.

---

### Decision · Program_Chairs · 2025-09-17

**Decision:**

Reject

**Comment:**

The paper proposes a prior-based data filtering method that estimates token priors from corpus-level term frequencies and filters documents using the mean and standard deviation of those priors, avoiding any model inference. It reports the highest average downstream performance across 21 benchmarks while reducing filtering time by >1000×, with demonstrated applicability to code, math, and dynamic multilingual corpora. Strengths include extreme efficiency and scalability, simplicity of deployment as a drop-in alternative to PPL, and clear empirical utility for large-scale pretraining pipelines. Potential weaknesses include unclear performance on harder benchmarks and problematic claims. In the rebuttal phase, both reviewers and authors actively participated in the discussion. By the end of the discussion phase, though most reviewers propose to accept the paper, reviewer wNnB raises unaddressed concerns and is not satisfied with the authors' response. After discussion with SAC, unfortunately we have to recommend rejection. I hope the authors could further revise this paper and submit to future venues.